# SimShear: Sim-to-Real Shear-based Tactile Servoing

**Yijiong Lin**[*], **Kipp McAdam Freud**[*], **Nathan F. Lepora**

School of Engineering Mathematics and Technology, University of Bristol

Bristol Robotics Laboratory, University of Bristol

Project webpage: https://yijionglin.github.io/simshear/

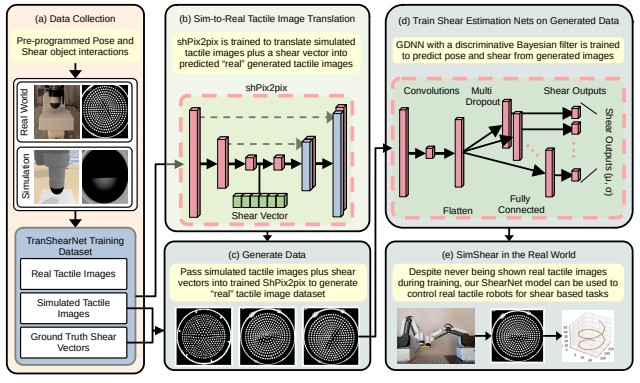

Figure 1. Overview of SimShear: our shear-based Sim-to-Real pipeline for tactile robotics.

## I. INTRODUCTION

This research aims to improve tactile sim-to-real policy transfer by leveraging contact data available in most rigid-body physics simulators to build sim-to-real image translators that include shear information and return estimated 'real' tactile images that correctly contain shear deformations. By transforming simulated images into their real analogs during training, it becomes possible to build policies that operate purely on real tactile images, thereby removing the need for a real-to-sim transformation at each deployment step. This streamlined process reduces computational overhead and simplifies the overall control pipeline. Our contributions are:

1. We introduce shPix2pix: a conditional U-Net GAN architecture that incorporates shear information into simulated tactile sensor images for image-to-image translation. By modeling deformations due to lateral displacements, shPix2pix enables the generation of realistic tactile images that contain shear deformations not modeled by our rigid-body simulator.

2. We train a ShearNet: a Gaussian Density Neural Network that leverages the shPix2pix-generated shear-based tactile images to estimate both contact pose and shear. We validate GDNN performance when trained on 'real' shPix2pix generated tactile images and show this significantly outcompetes a baseline using standard pix2pix-generated tactile images.

3. We demonstrate SimShear with two control tasks involving a pair of low-cost desktop robotic arms: a collaborative tactile tracking task and a collaborative co-lift task. Our results demonstrate how our shear-based sim-to-real approach enables manipulation using shear and validates that shear-aware models trained in simulation can effectively transfer to reality.

## II. METHOD

### A. Tactile Simulation

We use Tactile Gym 2.0 [1], a simulation environment designed specifically for tactile robotic manipulation tasks. As the simulated tactile sensor is approximated with rigid body physics, shear deformations of the tactile sensor are not modeled, and thus will never be present in simulated tactile images. However, we can infer shear displacements by extracting positional and rotational displacements from the simulator. When the sensor tip moves while in contact with an object, the shift in the contact surface's position and orientation relative to the sensor is used to generate a 6-dimensional "shear pose" (here 4 dimensional to match the DoFs of the robot arm), containing both positional and rotational shear components that complement the purely indentation depth-based images.

### B. Data Collection

The collected datasets consist of tuples containing real tactile images, corresponding simulated tactile images, and a vector encoding the shear displacements derived from the physics simulator. The robot performed a series of controlled interactions with the surfaces and edges of various objects, replicating a range of realistic contact scenarios.

### C. Conditional U-Net GAN for Image-to-Image Translation

Here we extend the vanilla pix2pix architecture to incorporate shear information explicitly. While previous methods [1], [2] used the standard U-net architecture to address the tactile sim-to-real gap, they did not account for shear information due to the limitations of rigid-body simulations. The U-net architecture cannot simulated tactile images into realistic ones due to the many-to-one relationship between simulated and real data. To address this, we add a fully connected layer with ReLU activation between the encoding (downsampling) and decoding (upsampling) layers of the U-net. The shear vector, which encodes both positional and rotational shear, is appended to the encoded representation before being passed through this fully connected layer.

### D. Training Generalizable Contact Pose Estimators

we extend this pose-and-shear decoding approach to simulated tactile images generated using our trained shPix2pix networks: the conditional U-Net GANs described in Figure 1. We trained Gaussian-density neural networks to decode both contact pose and shear information from our generated tactile images. By successfully training on simulated images, these

models can provide accurate pose and shear estimates on real tactile images despite the absence of direct real-world data during training.

Our Gaussian-density neural networks were trained using the ShPix2pix generated datasets. The training data included U-net generated tactile images and corresponding pose and shear vectors. The networks were optimized using a negative log-likelihood loss over 50 training epochs with batch size of 64, using the Adam optimizer [3] with a learning rate of 0.0001 and early stopping

## III. EXPERIMENTS AND RESULTS

### A. Task Formulation

We evaluate our improved Sim-to-Real pipeline using two collaborative tasks [4], [5] that require both pose and shear-based tactile sensing. In both scenarios, two Dobot MG400 robotic arms are employed: one acts as the leader robot, while the other (the follower robot) is equipped with a biomimetic optical tactile sensor that uses a trained GDNN shear-estimation network.

*a) Tactile Tracking Task::* In this task, the leader robot manipulates and rotates an object along a pre-programmed trajectory, and the follower robot actively tracks the object's surface while preserving continuous contact as it moves and rotates in three-dimensional space.

*b) Collaborative Co-Lifting Task::* Both robots hold the object together, with the leader robot moving it along a specified path while the follower robot maintains a secure grip.

### B. Sim-to-Real Image Translation

First, we evaluated our shPix2pix image translation network compared to a baseline vanilla pix2pix architecture. The vanilla pix2pix framework struggled to translate simulated tactile images into realistic ones due to the many-to-one relationship between simulated and real data. The baseline pix2pix architecture achieved a mean average pixel error (MAPE) of 0.22 and a structural similarity index measure (SSIM) of 0.17 when translating from simulated to real images.

Our shPix2pix image translation network successfully generated realistic tactile images that closely matched those obtained from the real tactile sensor. Explicitly encoding shear information into the model representation gives a significant reduction in MAPE to 0.091 and SSIM increase of 0.65.

### C. Sim-to-Real PoseNet

Our results indicate that while models trained on pix2pix-generated images can decode pose with above-random success, they are unable to infer shear information . In contrast, the models trained on shPix2pix images are able to accurately predict both pose and shear variables on real data never encountered during training. Note that the prediction errors for shear in the shPix2pix-trained model are comparable to those reported for networks trained entirely on real tactile data [6], highlighting the robustness of our approach.

### D. Tactile Object Tracking and Collaborative Co-Lifting Task

In the tactile object tracking task, we tested the tactile robot's ability to follow a leader robot across various complex trajectories. The follower MG400 successfully maintained continuous contact with the moving object over all tested trajectories. On average, the distance between the tactile sensor's position and the target position on the object's surface was 1-2 mm, as marked by the error on the trajectory plots. Repeated runs of the same experiment gave the same results, with the 'looping' trajectory most difficult because of the complex shape. Visually, the tracked trajectories of the tactile follower closely matched the trajectory of the leader robot (see supplemental video on the project webpage), confirming the precision of the sim-to-real tactile servo controller.

In the collaborative co-Lifting task, the tactile follower robot maintained continuous contact that was sufficiently precise in following the leader robot to securely hold the object while it was being moved along the leader' trajectory. Again, the trajectory errors were in the range 1-2 mm, consistent with a close visual match between the leader and follower trajectories (see supplemental videos on the project webpage). The most challenging object was the soft brain, because the soft contact led to deformation of the tactile images and less reactive control, although good performance was still maintained.

## IV. CONCLUSION

We introduced SimShear, a novel Sim-to-Real pipeline for tactile sensing that integrates shear information to enhance the accuracy and versatility of robotic manipulation tasks trained in simulation. By employing a conditional U-Net GAN, the shPix2pix, our method overcomes the limitations of current Real-to-Sim pipelines that cannot model the effects of shear force. Our method removes two primary drawbacks of current tactile real-to-sim pipelines—namely, the lack of shear in simulation and the need to translate real images into the simulated domain at every inference step. Instead, we generate realistic, shear-enabled tactile data for policy training, allowing robots trained in simulation to sense and respond to lateral displacements directly in real-world scenarios.

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
