# OpenReview forum: "SimShear: Sim-to-Real Shear-based Tactile Servoing"
_IEEE.org/IROS/2025/Workshop/Tactile_Sensing — IROS 2025 Workshop Tactile Sensing Poster_

### Official Review · Reviewer_JyZS · 2025-09-15
**Good paper, a few concerns**

**Rating:** 6
**Confidence:** 3

**Review:**

The paper introduces a generative model that modifies from previous design by including shear information to better translate the simulated sensor to a real one. Paper is overall well-written. Experimental setup is well explained, and the necessity for this task is clear however there are a few concerns/minor points on the paper.

1. The author compared it to the pix2pix, but not stated how many trials the MAPE is averaged over. This would help determine whether shear information is helping the model with significance. When the author sates it gave significant reduction, did they use significance testing methods to evaluate this? If so this should be stated in the results section.

2. How does the authors model compare to other sensors? According to https://ieeexplore.ieee.org/abstract/document/9811801 the authors model model also outperforms the gelsight using pix2pix. Could the authors method be useful to other sensors, not just the TacTip? Some sort of table showing existing pix2pix methods for tactile sensing and model performance could highlight significance of the authors model pipeline.

3. Is the MAPE the standard metric? Other papers have used mean absolute error (MAE)

4. Has the author published the datasets used? The work cannot be replicated or validated if the datasets are not published. For a workshop this is not necessary but if one intends to publish this work later on they should consider doing this.

---

### Official Review · Reviewer_449L · 2025-09-22
**Relevant idea, presentation could be improved**

**Rating:** 7
**Confidence:** 5

**Review:**

The paper describes how to augment domain-adaptation for tactile sensing with accurate shear force simulation. The problem is relevant as shear forces are crucial for slippage control. However, the paper would benefit of an experiment that actually shows that training a policy with pressure+shear information is better than one trained only with pressure information. I am not fully convinced that tactile servoing is the ideal task for this ablation.

Also, the way experiments are presented is not very clear. I understand the length limitations, but I think it would be beneficial to replace some text with a figure or a table that better summarizes the experimental results.